# C-Reactive Protein as a Possible Predictor of Trail-Making Performance in Individuals with Psychiatric Disorders

**DOI:** 10.3390/nu12103019

**Published:** 2020-10-02

**Authors:** Nina Dalkner, Eva Reininghaus, Karin Schwalsberger, Alexandra Rieger, Carlo Hamm, René Pilz, Melanie Lenger, Robert Queissner, Valerie S. Falzberger, Martina Platzer, Frederike T. Fellendorf, Armin Birner, Susanne A. Bengesser, Elisabeth M. Weiss, Roger S. McIntyre, Harald Mangge, Bernd Reininghaus

**Affiliations:** 1Department of Psychiatry and Psychotherapeutic Medicine, Medical University Graz, 8036 Graz, Austria; nina.dalkner@medunigraz.at (N.D.); carlo.hamm@kages.at (C.H.); rene.pilz@medunigraz.at (R.P.); melanie.lenger@medunigraz.at (M.L.); robert.queissner@medunigraz.at (R.Q.); valerie.falzberger@gmail.com (V.S.F.); martina.platzer@medunigraz.at (M.P.); frederike.fellendorf@medunigraz.at (F.T.F.); armin.birner@medunigraz.at (A.B.); susanne.bengesser@medunigraz.at (S.A.B.); 2Therapiezentrum Justuspark, 4540 Bad Hall, Austria; karin.riedrich@gmail.com (K.S.); alexandra.rieger@medunigraz.at (A.R.); berndreininghaus@web.de (B.R.); 3Institute of Psychology, University of Innsbruck, 6020 Innsbruck, Austria; Elisabeth.Weiss@uibk.ac.at; 4Department of Psychiatry and Pharmacology, University of Toronto, Toronto, ON 399, Canada; roger.mcintyre@uhn.on.ca; 5Research Unit on Lifestyle and Inflammation-associated Risk Biomarkers, Clinical Institute of Medical and Chemical Laboratory Diagnostics, Medical University, 8036 Graz, Austria; harald.mangge@medunigraz.at; 6BioTechMed-Graz, 8036 Graz, Austria

**Keywords:** inflammation, C-reactive protein, obesity, cognition, psychomotor processing speed, cognitive flexibility, Trail-Making Test

## Abstract

Cognitive dysfunction is a prominent feature of psychiatric disorders. Studies have shown that systemic low-grade inflammation is crucial in the development of cognitive deficits across psychiatric disorders. The aim of this study was to further examine the role of inflammation and inflammatory mediators in cognitive function in psychiatric disorders. This study included 364 inpatients (53% females) with International Classification of Diseases (ICD)-10 F3 (affective disorders) and F4 (neurotic, stress-related, and somatoform disorders) diagnoses. The mean age was 52 years (22 to 69 years) and the median body mass index was 27.6. Cognitive function was assessed with the Color–Word Interference Test after Stroop and the Trail-Making Test A/B. Multiple linear regression models were calculated to assess the predictive value of C-reactive protein and the kynurenine/tryptophan ratio on cognitive function controlling for age, sex, education, premorbid verbal intelligence quotient, illness duration, depressive symptoms, and obesity-related parameters (e.g., body mass index, high-density lipoprotein). Our data confirm that in patients with psychiatric disorders, C-reactive protein serum concentration is a relevant and important predictor of Trail-Making Test B performance, measuring cognitive flexibility. The effect size of this association did not change much after adding clinical and metabolic variables into the regression model. The kynurenine/tryptophan ratio was not related to cognitive test scores. The involvement of C-reactive protein as a peripheral inflammatory marker in cognitive flexibility and psychomotor processing speed in psychiatric illness can be concluded.

## 1. Introduction

Cognitive impairment, which is a prominent core feature across psychiatric disorders [1,2,3,4,5,6] includes deficits in attention, short- and long-term memory, understanding, creativity, knowledge, word meaning, verbal fluency, receptive vocabulary, psychomotor speed, problem solving, planning, reasoning, and judgment. The latter are known as executive functions. Executive functioning generally refers to “higher-level” cognitive function and is thought to be subserved by the prefrontal cortex [7]. Cognitive deficits decrease psychosocial functioning, aggravate the course of psychiatric disorders, and are strongly related to patients’ quality of life and well-being [8,9]. Recent studies focus on the biological underpinnings of cognition in psychiatric disorders including changes in neurotransmission at the synapse, homeostatic synaptic plasticity, and altered activation of various pathways such as the hypothalamic–pituitary–adrenal (HPA) axis and the kynurenine (KYN) pathway, which is suggested to be involved in learning and memory across psychiatric disorders [10,11]. However, the biological mechanisms underlying cognitive deficits in psychiatric disorders are not fully understood.

Neuroinflammation is suggested to play a key role in cognitive deficits in psychiatric disorders. It is well known that immune dysregulation and elevated levels of inflammatory cytokines often accompany major psychiatric disorders [12,13,14]. In a meta-analysis, pro-inflammatory cytokines were revealed to contribute to the causal mechanism in mental disorders [15]. Accordingly, obesity and metabolic disturbances such as dyslipidemia are major risk factors for medical morbidity and mortality in patients with psychiatric disorders [16,17].

High body mass index (BMI) is associated with a severe course of psychiatric disease and specific neurostructural alterations (i.e., gray matter reductions) [18]. It is suggested that inflammation could be understood as a pathophysiological nexus between mood disorders and obesity [19]. Consequently, elevated levels of high-sensitive C-reactive protein (hsCRP), which is the most common and robust marker of systemic inflammation and predictor of cardiovascular disease, have been associated with anthropometric measures of obesity [20], especially central obesity [21], major depression [13,22], and also cognitive function [23,24]. Obesity is characterized by the activation of inflammatory processes in metabolically active sites such as adipose tissue, liver, and immune cells [25,26]. Chronic inflammation in adipose tissue and oxidative stress contributes to the development of insulin resistance and ends up in obesity-related disorders (i.e., diabetes mellitus, atherosclerosis, hyperlipidemias) [26,27].

In addition, obese subjects were found to have impaired executive function, especially in decision-making tasks, attention, and performance during the Stroop Test [28]. In one study, the risk of executive function deficits has been found to be four times higher in obese individuals than in non-obese individuals, independently of their demographic and medical characteristics [29]. It has been suggested that executive function moderates the intention–behavior link for food intake and physical activity [30,31]. In sum, executive function deficits are related to poor psychosocial functioning and disability in patients with major depressive disorder [32]. Cognitive deficits may be associated with both, low-grade inflammation, which may represent the earliest consequence of obesity, as well as other metabolic disturbances including insulin resistance resulting in diabetes mellitus type 2, elevated lipids, and hypertension. However, weight problems might also precede cognitive deficits in psychiatric patients, as cognitive impairment leads to a poor lifestyle, which has been associated with weight gain [26]. The mechanisms via which obesity-induced inflammation affects cognition have not yet been fully explained.

Tryptophan (TRP) is an essential amino acid and crucial in serotonin synthesis. The TRP-metabolite KYN is produced by the action of a largely hepatic-based enzyme tryptophan-2,3-dioxygenase (TDO) or the ubiquitous indoleamine-2,3-dioxygenase (IDO), particularly under pro-inflammatory conditions and activation of T-lymphocytes. A disturbance in TRP metabolism has been linked to cardiovascular disease [33], and the KYN to TRP ratio (KYN/TRP), an estimate of TRP breakdown, has been associated with conditions of immune-mediated inflammation [34]. The KYN/TRP ratio as a parameter of bipolar disorder pathogenesis has been linked to obesity and was described in detail in our recent report [35]. Plasma KYN and neurotoxic KYN metabolites such as hydroxykynurenine were elevated in individuals with major depression, who had attempted suicide [36]. These metabolites have been proposed to be inflammatory mediators, have been linked to the pathogenesis of several neurodegenerative disorders, and have been found to be increased in bipolar subjects compared to those in controls [37].

High concentrations of hsCRP were associated with cognitive decline in the general population [38,39], and hsCRP was identified as risk factor for dementia, Alzheimer disease, and vascular dementia in the Rotterdam study [40]. Furthermore, hsCRP was negatively related to executive function, spatial reasoning, and processing speed in a nonclinical sample of elderly adults [23]. Yaffe et al. found that elevated serum levels of hsCRP were associated with cognitive decline in individuals with metabolic syndrome [41]. In a study by Sweat et al. [42], hsCRP was linked to lower performance in cognitive tests of frontal lobe function in overweight and obese women. A recent review by Misiak et al. showed that, in individuals with schizophrenia and bipolar disorders, hsCRP levels and cytokine alterations were significantly associated with worse cognitive performance in abstract reasoning, memory, learning abilities, attention, mental flexibility, and processing speed [43]. A negative relationship between hsCRP and performance in executive function tasks was recently observed in healthy individuals. However, associations between hsCRP and poorer executive cognitive functioning covaried with age and were not present after adjustment for demographical factors [44]. In a 12-year longitudinal study, pro-inflammatory cytokines predicted cognitive symptoms of depression; in contrast, cognitive symptoms at baseline did not predict hsCRP at follow-up [45]. A separate study conducted in patients with schizophrenia found that hsCRP levels did not predict longitudinal changes in cognitive domains including memory, visuospatial/constructional performance language, attention, and delayed memory over a course of three years [46]. Other studies could recently prove the relationship between elevated levels of hsCRP and performance in the Wisconsin Card Sorting Test, an executive function test [47], and psychomotor speed [48] in depressive individuals. However, the precise role of peripheral inflammation on cognitive symptoms in psychiatric disorders remains to be clarified. Because of methodological heterogeneity between studies, insufficient standardization, and a lack of control for confounds, more research with large samples controlling for relevant clinical and metabolic confounders is needed [43,49].

This study aims to broaden existing knowledge on the potential role of hsCRP and TRP and KYN as mediators of immune activation in cognitive functioning in a large sample undergoing intensive psychiatric rehabilitation care. In accordance with the knowledge that obesity, inflammation, and cognitive deficits are related, we assumed that hsCRP as a sensitive marker for systemic low-grade inflammation could be associated with poor performance in cognitive tests, and that the putative impairment is specific for some cognitive domains. We wanted to test the impact of obesity and metabolic disturbances on this association by controlling for relevant confounding variables such as BMI, high-density lipoprotein (HDL), triglycerides, blood pressure, and history of diabetes mellitus. Moreover, in contrast to some previous studies, a strict control of potential clinical and demographic confounding variables (e.g., illness variables, verbal intelligence quotient (IQ), depressive symptoms) was intended. In addition, we aimed to test the connection between performance in executive tasks and parameters of the KYN pathway, as KYN pathway metabolites were recently related to learning and memory [10,50], and also to tasks sensitive to frontal executive function [51].

## 2. Materials and Methods

A total of 364 patients diagnosed with a primary diagnosis of an affective disorder (ICD-10: F3) or a neurotic, stress-related, and somatoform disorder (ICD-10: F4) were included in the study. Diagnoses were assessed by specialists in psychiatry according to the International Classification of Diseases (ICD-10) [52]. All participants were treated at a psychiatric rehabilitation center in Austria between April 2015 and February 2016. All measures were conducted in the first treatment week. Almost half of the individuals were prescribed selective serotonin reuptake inhibitors(SSRIs), a smaller group took atypical antipsychotics, and 1.1% took lithium as a medication at the time of the study. Almost all patients were treated with combinations of drugs.

The patients undergoing the psychiatric rehabilitation program differed in their stage of recovery. In general, the psychiatric rehabilitation setting follows acute psychiatric inpatient treatment protocols; however, waiting times for therapy program initiation are different and dependent on insurance approval. Data were extracted from a large ongoing study measuring the neurobiological foundation of the burnout syndrome in psychiatric rehabilitation treatment. Exclusion criteria were main diagnosis with schizophrenic disorders (ICD-10: F2), neurodegenerative disorders, or mental retardation according to the study protocol. Patients above 70 years were excluded from the analyses of this study. All participants signed informed consent; the study was approved by the Ethics Committee of the Federal State, Upper Austria, EK-number: E-24-14.

### 2.1. Cognitive Tests

A cognitive test battery was administered by clinical psychologists measuring neuropsychological parameters indexing attention, psychomotor processing speed, executive function, as well as verbal IQ.

The Color–Word Interference Test after Stroop [53] is a neuropsychological tool to measure selective attention, visual processing, and cognitive flexibility. While the first two tasks of the Stroop Test (Stroop word reading, Stroop color naming) measure cognitive processing speed and attentional capacity, the third task (Stroop interference) measures response inhibition and response selection, indices of executive control. In this task, the subjects have to identify the color that a color word is printed in; however, trials are incongruent (e.g., “red” written in blue ink).

Another neuropsychological tool measuring attention, processing speed, and executive functioning is the Trail-Making Test A/B (TMT A/B) consisting of two test parts [54]. In test part B, subjects are instructed to alternately connect letters and numbers in sequence (A–1–B–2, etc.), while in test part A, they have to connect numbers only, which does not require set-shifting. While TMT A assesses psychomotor speed, visual scanning, and attention, TMT B measures cognitive flexibility and executive functioning.

The Multiple-Choice Vocabulary Test (MWT) was used to estimate the premorbid verbal intelligence level [55]. This verbal test has been shown to be valid to measure the premorbid ability, assessing knowledge, especially vocabulary.

### 2.2. Clinical Inventories and Parameters

The Beck Depression Inventory-2 was used as a self-report measure to assess depressive symptoms (BDI-2) [56]. External rating scales (Hamilton Depression Scale (HAMD)) [57] and Clinical Global Impressions (CGI) [58] were conducted by a trained clinician. BMI, systolic/diastolic blood pressure, and other anamnestic (i.e., history of diabetes, illness duration) as well as demographic data (i.e., education) were collected by trained clinic staff.

### 2.3. Biological Assays

Fasting serum was collected between 8 and 9 a.m. and either processed immediately for further analyses (metabolic parameters, hsCRP) or stored at −80 °C until analysis. The metabolic parameters of HDL, low-density lipoprotein (LDL), and triglycerides were analyzed by Labcon, Austria. Levels of hsCRP were analyzed by Abbott GmbH with Architect ci8200. Free TRP and KYN serum concentrations were determined by high-performance liquid chromatography [59]. The KYN/TRP ratio was calculated as an index of TRP breakdown, thereby providing a proxy of IDO activity.

### 2.4. Statistics

The strength of the associations between cognitive function and hsCRP was assessed by linear regression analyses. Analyses were first run for hsCRP alone (Model 1), then adjusted for age, sex, premorbid IQ, illness duration, and Beck Depression Inventory-2 (Model 2), and then additionally for metabolic parameters (current levels of HDL, triglycerides, diastolic blood pressure, history of diabetes, and BMI (Model 3)). These variables were chosen for the analyses based on their known association with inflammation and cognitive function as well as the association with hsCRP or cognition in the present study (*p* < 0.05). To avoid co-linearity, highly correlated variables (r > 0.50) were not allowed in the models. Therefore, LDL cholesterol (correlated significantly with HDL cholesterol), systolic blood pressure (correlated significantly with diastolic blood pressure), and education (correlated significantly with verbal IQ) were excluded of calculating. As associations were found between TRP and KYN as well as between TRP and KYN with hsCRP, we only used the KYN/TRP ratio and ran a second regression analysis to avoid co-linearity for assessing the relationship between the KYN/TRP ratio and cognitive function. Basic statistical requirements to run linear regression analyses were met, and preliminary analyses were conducted to check relevant assumptions of regression analysis including linearity, normality, absence of multicollinearity, and homoscedasticity. The distribution of hsCRP and triglycerides was highly skewed. Skewed variables were transformed by log transformation or inverse transformation and outliers were removed. The overall significance was set at 0.05 and the Simes’ test and the Benjamini–Hochberg false discovery rate (FDR) procedure correcting for multiple comparisons was applied [60]. Data were analyzed using IBM SPSS for Windows, version 23.

## 3. Results

### 3.1. Description of the Study Population

Fifty-five percent of the participants were female. Eighty-four (23.1%) participants were diagnosed with anxiety, dissociative, somatoform, and other nonpsychotic mental disorders (ICD-10: F4), 26 (7.14%) were diagnosed with bipolar disorder (ICD-10: F31), 93 (25.6%) were diagnosed with depressive episodes (ICD-10: F32), 122 (35.5%) were diagnosed with recurrent depressive disorder (ICD-10: F33), and 39 (10.7%) had a comorbidity of anxiety and depression (ICD-10 F3 and F4).

Table 1 summarizes clinical data (BDI-2, HAMD, CGI, illness duration) as well as biological variables (BMI, blood pressure) and biomarkers (hsCRP, HDL, LDL, triglycerides, TRP, KYN, and KYN/TRP ratio). The obesity rate (BMI ≥ 30) in the sample was 20.9%, the overweight rate (BMI ≥ 25) was 37.1%, the diabetes rate was 7.7%, the cigarette smoking rate was 25.5%, and the rate of problematic alcohol consumption (more than 5 alcoholic drinks/2–3 times per week) was 6.2%. Thirty patients reported having had cancer, and 63 patients reported having had or having a cardiovascular disease; however, this information was based on self-reports and unspecified and, therefore, not used for further analyses. Fifty-four percent of the participants exercised regularly (more than 1 time/week). Of the individuals included, 60.4% completed high school or higher education (e.g., university). The mean verbal IQ was 112.4 (SD = 14.0). Table 2 presents the cognitive variables.

HsCRP was related to BMI (r = 0.49, *p* < 0.001), HDL (r = −0.24, *p* < 0.001), triglycerides (r = 0.28, *p* < 0.001), systolic blood pressure (r = 0.15, *p* < 0.01), diastolic blood pressure (r = 0.22, *p* < 0.001), KYN (r = 0.14, *p* < 0.05), KYN (r = 0.18, *p* < 0.01), and KYN/TRP ratio (r = 0.19, *p* < 0.01).

We used one-sample t-tests to compare cognitive test scores in the sample with norm values from the manuals [52,53,54]. Participants had deficits in all analyzed cognitive tasks (*p* < 0.001).

### 3.2. Results from Multiple Regression Analyses

#### 3.2.1. Associations between hsCRP and Cognitive Function

Table 3 gives the results of the linear regression analyses using TMT A and TMT B as dependent variables and hsCRP as independent variable (Model 1). Regression models were then adjusted for age, sex, verbal IQ, illness duration, and BDI-2 (Model 2), and then additionally for relevant obesity-related variables and biomarkers including BMI, diastolic blood pressure, triglycerides, HDL cholesterol, history of diabetes (Model 3). In the tables, standardized regression coefficients, t-values, and *p*-values for each parameter are listed. Table 4 shows the results of the regression analyses for the Stroop parameters (Stroop word reading, Stroop color naming, Stroop interference) as the dependent variable, and hsCRP as independent variable.

First, it has been shown that hsCRP predicted cognitive performance in the TMT B (see Figure 1). Controlling for relevant confounders weakened the correlation, indicating a trend after the FDR correction procedure. Age and verbal IQ were highly related to TMT A/B performance (see Table 3). In addition, BMI and BDI-2 were significant associated with TMT A performance.

Second, hsCRP did not predict Stroop performance. Age and IQ were significant confounders in all three Stroop tasks, and BDI-2 was associated with performance only in the Stroop interference task.

None of the biomarkers or metabolic parameters (HDL, triglycerides, blood pressure, history of diabetes) nor sex or illness duration were significantly associated with cognitive variables in this data set.

#### 3.2.2. Associations between KYN/TRP Ratio and Cognitive Function

Linear regression analyses showed that the KYN/TRP ratio was no significant predictor of cognitive performance in this sample (TMT A: β = 0.07, t = 1.28, *p* = 0.203; TMT A: β = 0.09, t = 1.16, *p* = 0.111; Stroop word reading: β = 0.04, t = 801, *p* = 0.423; Stroop color naming: β = 0.07, t = 1.19, *p* = 0.236; Stroop interference: β = 0.08, t = 1.41, *p* = 0.159).

## 4. Discussion

The purpose of this study was to explore the potential role of inflammatory processes under control of clinical and obesity-related parameters on cognitive task performance in individuals with a lifetime psychiatric affective or anxiety/somatoform disorder undergoing psychiatric rehabilitation. We hypothesized that the inflammatory marker hsCRP might contribute to deficits in psychomotor speech and executive function in psychiatric patients. In addition, the tryptophan–kynurenine metabolism, especially the KYN/TRP ratio, was suggested to play a role in cognitive performance. Results further elucidated the role of hsCRP in psychomotor speech and executive function in psychiatric patients. However, there was no evidence to support the role of TRP metabolites. The major strengths of this study include the sample size, the acquisition of all data in a single center, and the use of various clinical variables. The incorporation of multiple covariates (e.g., illness variables, demographics, verbal IQ, metabolic variables) in the regression model further strengthens the results.

The main finding was that hsCRP levels were associated with poorer performance in TMT B. Notably, this effect did not change when covariates were added (from β = 0.14 to β = 0.10–0.12). However, the significance of the association between hsCRP and TMT B decreased in a regression model with all variables and did not survive a significance level that was corrected for multiple comparisons.

Prefrontal cortical activation during TMT B performance has been observed earlier, which may reflect cognitive set shifting [61], which is an executive function that involves the ability to unconsciously shift attention between one task and another. Our finding is in accordance with data from previous studies in psychiatric samples linking high hsCRP concentrations to trail-making performance and executive functioning [43,48]. Executive functioning was linked to hsCRP in the large-scaled Rotterdam Study [39] as well as in the Berlin Aging Study II [23]. Consistently with our results, hsCRP was moderately associated with executive function and decline in the elderly in those studies. The authors concluded that systemic markers of inflammation may not be suitable for risk stratification [39]. Moreover, in line with our results are the recent findings by Stenfors et al. showing a relationship between inflammation, especially hsCRP, and executive function [44]. However, Stenfors et al. observed an association between inflammatory processes and cognition only among participants without reported health conditions (e.g., diabetes). The authors concluded that the relationship might be due to other factors involved in conditions that affect cognition (e.g., affectivity) and hsCRP (e.g., vascular risk factors). Indeed, Stenfors et al. included the self-reported presence/absence of cardiovascular disease, high blood pressure, and diabetes in their analyses. In contrast to our study, they did not test the impact of metabolic biomarkers. Finally, our results show that obesity-related variables and metabolic biomarkers were less important in the association between hsCRP and TMT B performance. Nevertheless, future studies have to clarify the true role of obesity-induced chronic inflammation on aspects of cognitive function in detail.

Interestingly, we did not find any association between Stroop interference and hsCRP. Although both tasks, TMT B and Stroop interference task, measure executive function and response inhibition, studies conclude that trail-making tests are most suitable for assessing processing speed and cognitive flexibility more than the ability to maintain a complex response set [62]. However, both tasks require other cognitive demands, including visual search, motor speed, sequencing, and working memory for task rule [5]. Demakis suggests that Stroop interference seems to be a bit more sensitive for measuring frontal lobe damage [63]. Sweat et al. also found that obesity-associated inflammation appears to be associated with cognitive dysfunction in frontal lobe tasks; however, effects were found only in females, and results were driven by the overweight/obese female group [42]. In our study, BMI was related to TMT A performance measuring psychomotor speed, visual scanning, and attention. Evaluating sex and obesity effects in psychiatric samples could be promising for future research. Although the median BMI in our sample was in the overweight range, only 28% were obese. Analyzing a primary obese sample (median BMI ≥ 30) could shed further light on the effects of obesity on executive functioning and the impact of these association on inflammation and vice versa.

The KYN/TRP ratio, as an indicator of TRP breakdown has been considered to reflect specific inflammatory pathways as it is frequently used to express or reflect the IDO activity. Recent studies found that the levels of TRP, KYN, and the KYN/TRP ratio were significant predictors of cognitive performance in tasks sensitive to frontal executive function and memory [10,51]. Thus, TRP catabolites (e.g., kynurenine acid, 3-hydroxikynurenine) have recently been related to worse memory function in men with bipolar disorder [50]. Against our assumption, the KYN/TRP ratio did not correlate with executive function in this study. This leads us to the conclusion that KYN and TRP play a minor role in executive functioning as a symptom in psychiatric diseases. Accordingly, other studies also failed to prove the mediating role of KYN/TRP ratio on symptoms of affective disorders [64]. However, more studies in this context are needed.

### 4.1. Clinical Impact

This is one of the first studies investigating a large psychiatric sample regarding the relationship between inflammation and cognitive function by adding numerous clinical variables, which were included in a regression model. This approach further broadens our knowledge of the influence of inflammatory mediators on cognitive impairment in psychiatric disorders and the role of clinical and metabolic variables in this context. Age and verbal IQ are the most significant variables contributing to cognitive deficits in psychiatric patients. In addition, current depressive symptoms measured with the BDI-2 were also highly related to attentional function (TMT A) and Stroop interference performance but less to cognitive flexibility (TMT B), Stroop word reading, and Stroop color naming. What is important for clinicians is that hsCRP could be used as a potential predictor for a cognitive flexibility task, the TMT B. In addition, data confirm that hsCRP is a better predictor of cognitive impairment than metabolic biomarkers (HDL/triglycerides) or obesity-related variables (blood pressure and history of diabetes mellitus) are. Nevertheless, as parameters of overweight and obesity are common correlates of systemic inflammation [19,20] and hsCRP was related to BMI and metabolic parameters also in this study, lifestyle interventions to lose weight should be included in the treatment of anxiety and mood disorders to a greater extent. Individuals in weight loss trials experienced reductions in symptoms of depression [65] as well as in cognitive impairment [66]. Recently, numerous research efforts have reported a beneficial effect of nutrients such as n-3 polyunsaturated fatty acids and flavonoids on inflammatory processes underlying cognitive dysfunctions [67]. Similar findings were reported for a brief lifestyle intervention, including a vegan diet rich in fresh fruits and vegetables, whole grains, and various legumes, nuts, and seeds, to reduce systemic inflammation as measured by circulating hsCRP [68].

Our research further elucidates the mechanisms underlying cognitive deficits in psychiatric disorders and provides evidence to inform the development of multidisciplinary approaches to ameliorate cognitive deficits in clinical practice. Training programs for psychiatric patients such as cognitive remediation of neuropsychological deficits [69], as well as conventional psychoeducational intervention programs for psychiatric patients should additionally include scientific knowledge about the harmful effects of systemic low-grade inflammation affecting patient’s physical and psychological health [13,14,15,19,21,22,23]. Additionally, more research would be desirable regarding the effects of dietary management to alleviate cognitive impairment and to investigate whether those cognitive improvements are mediated by inflammatory changes in psychiatric patients.

### 4.2. Limitations

This study has certain limitations. First, we could not identify causal directions with the cross-sectional study design and, therefore, reverse causation could be a possibility. Poor cognitive abilities could have provoked unfavorable behaviors, influencing inflammatory processes. It has to be clarified in future studies whether the cognitive decline in individuals with psychiatric disorders may be an indirect consequence of metabolic disturbances, due to insulin resistance, or a direct effect of obesity-provoked chronic low-grade inflammation.

Second, another limitation was the heterogeneity of the sample concerning psychiatric symptomatology and illness duration; however, we attempted to control for these variables in the analyses. Pro-inflammatory cytokines seem to play an important role in the clinical course of psychiatric disorders, and it is suggested that inflammatory markers can be associated with the severity and quantity of depressive symptoms [12,14]. As inflammatory processes seem to be mediated by symptom severity in affective disorders, the high heterogeneity of the sample regarding depression scores and systematic inflammation may have influenced the results. However, our results indicate that performance in cognitive flexibility tasks (TMT B) seems to be far more sensitive to low-grade inflammation measured by hsCRP than performance in pure attentional tasks. Due to the high number of investigated patients in our study, there is a strong power for this interaction.

Third, we did not have a control group and, therefore, it remains unclear whether the findings are specific to psychiatric patients or not. Furthermore, the sample was limited to individuals with mood and anxiety disorders, which reduces the generalizability of the results to other diagnostic groups. However, we expect associations to be similar across different psychiatric groups as a recent review summarized evidence that supports possible shared mechanisms including inflammatory processes in the establishment of cognitive impairments across different psychiatric conditions. Fourrier et al. therefore suggest considering a cross-disorder approach including pharmacological and nonpharmacological (i.e., physical activity and nutrition) anti-inflammatory/immunomodulatory strategies in the management of cognition in psychiatry [11].

Fourth, although relevant somatic conditions were controlled for (e.g., diabetes, hypertension, and dyslipidemia), it cannot be said for sure that all subjects were free of medical comorbidities (cancer, cardiovascular disease, etc.).

Fifth, medication, which could influence cognitive functioning, was not involved in the analyses due to potential medication combinations. Controlling for every medication would have yielded a myriad of permutations of psychopharmacological treatment combinations that do not fit for statistical analysis.

Sixth, we do not know for sure whether the mechanism of inflammation is similar or different between patients with affective disorders (unipolar versus bipolar disorder) and those with neurotic, stress-related, and somatoform disorders. Because of the high comorbidity, it was not possible to split these groups. Future studies should analyze homogenous groups regarding psychiatric diagnoses. Furthermore, additional serum biomarker such as IL-1 or IL-6 might be of interest to test the influence of peripheral inflammation on cognitive dysfunction.

## 5. Conclusions

This study underscores the relevance of hsCRP, an important inflammatory marker, in predicting cognitive function in patients with mood- and anxiety-related disorders. Particularly, hsCRP concentration was a significant predictor of poor cognitive flexibility measured with the Trail-Making Test. While the influence of clinical factors (e.g., age, verbal IQ, depressive symptoms) on trail-making performance was strong, hsCRP was a significant predictor of trail-making performance. We conclude that cognitive flexibility (measured with the TMT B) seems to be especially sensitive for low-grade inflammatory influences. Future longitudinal research is needed to test whether there are causal relationships between inflammatory parameters and cognitive dysfunction in different psychiatric samples. The elaboration of more specific treatment strategies (e.g., psychopharmacological or lifestyle interventions) to regulate inflammation in psychiatric disorders would be promising.

## Figures and Tables

**Figure 1 nutrients-12-03019-f001:**
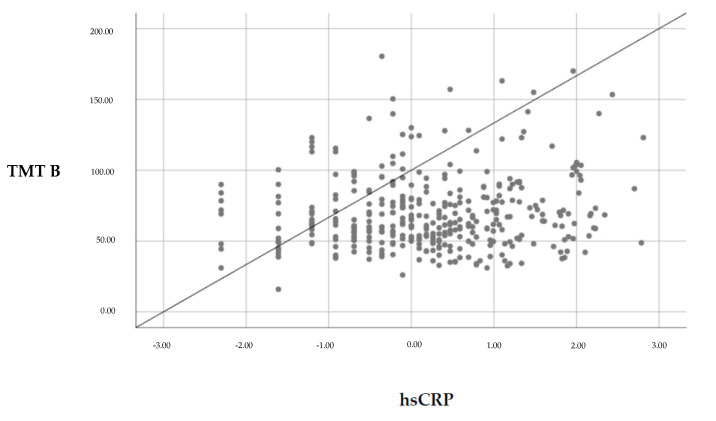
Association of serum high-sensitive C-reactive protein (hsCRP) concentration and Trail-Making Test B performance.

**Table 1 nutrients-12-03019-t001:** Demographic, clinical, and biochemical characteristics (mean, standard deviations, minimum, maximum; *N* = 364).

	*N*	*M*	*SD*	*Min*	*Max*
**Demographics and clinical characteristics**					
Age [years]	364	52.4	7.5	22.8	67.3
Verbal IQ	346	112.4	14.3	86	145
Illness duration [years]	364	12.4	12.1	0.1	55.5
**Psychopathology**					
HAMD	327	10.8	6.6	0	31
BDI-2	358	20.1	9.8	0	51
CGI	357	3.6	0.9	0	6
**Metabolic risk variables**					
Systolic blood pressure [mmHg]	361	141.4	18.6	98	198
Diastolic blood pressure [mmHg]	361	88.0	11.6	55	129
BMI [kg/m^2^]	360	26.6	4.7	18.5	46.5
**Biomarkers**					
hsCRP [mg/L]	362	3.34	2.5	0.01	16.6
TRP [ng/mL]	355	65.5	8.4	38.9	92.7
KYN [ng/mL]	355	1.9	0.46	0.94	4.4
KYN/TRP	355	0.03	0.01	0.02	0.07
LDL [mg/dL]	363	144.4	38.1	45	250
HDL [mg/dL]	363	54.6	15.4	26	104
Triglycerides [mg/dL]	364	141.9	76.0	38	537

Note: HAMD = Hamilton Depression Scale, BDI-2 = Beck Depression Inventory-2, CGI = Clinical Global Impression, LDL = low-density lipoprotein, HDL = high-density lipoprotein, BMI = body mass index, hsCRP = high-sensitive C-reactive protein, TRP = tryptophan, KYN = kynurenine, KYN/TRP = kynurenine/tryptophan ratio.

**Table 2 nutrients-12-03019-t002:** Cognitive test scores (mean, standard deviations, minimum, maximum).

		*N*	*M*	*SD*	*Min*	*Max*
**Cognitive tasks**						
*Verbal IQ*		364	112.4	14.3	86	145
*Attention*						
	TMT A [s]	364	33.4	14.0	9.9	120.4
	Stroop word reading [s]	364	31.3	8.3	18	138
	Stroop color naming [s]	364	47.2	10.0	26	111
*Cognitive flexibility/Executive function*						
	TMT B [s]	364	70.0	27.4	15.9	223
	Stroop interference [s]	364	112.4	14.3	41	185

Note: TMT A = Trail-Making Test A, TMT B = Trail-Making Test B.

**Table 3 nutrients-12-03019-t003:** The association of serum hsCRP concentration with linear multiple regression on Trail-Making Test performance (*N* = 335).

	TMT A			TMT B		
	**β**	***t***	***p***	**β**	***t***	***p***
hsCRP ^a^	0.1	1.79	0.074	0.14	2.52	0.012 *
hsCRP ^b^	0.06	1.27	0.207	0.1	2.09	0.037
Age	0.37	7.07	<0.001 *	0.4	7.78	<0.001 *
IQ	−0.18	−3.52	<0.001 *	−0.23	−4.55	<0.001 *
Sex	0.01	0.172	0.836	0.12	2.38	0.018
Illness duration	−0.04	−0.84	0.006 *	0.03	0.667	0.505
BDI-2	0.14	2.67	0.008 *	0.06	1.19	0.234
hsCRP ^c^	0.13	2.19	0.029	0.12	2.15	0.033
Age	0.4	7.32	<0.001 *	0.41	7.76	<0.001 *
IQ	−0.20	−3.79	<0.001 *	−0.24	−4.70	<0.001 *
Sex	0	0	0.999	0.12	2.1	0.037
Illness duration	−0.05	−0.87	0.387	0.04	1.18	0.495
BDI-2	0.14	2.77	0.006 *	0.06	0.68	0.238
BMI	−0.17	−2.65	0.008 *	−0.07	−1.04	0.301
DBP	0	0.07	0.945	0.02	0.28	0.783
Triglycerides	0.03	0.44	0.66	0.07	1.11	0.267
HDL	−0.05	−0.73	0.467	0.03	0.38	0.706
Diabetes	−0.01	−0.09	0.926	−0.06	−1.1	0.263

Note: Data of linear regression analyses for high-sensitive c-reactive protein (hsCRP) and Trail-Making Test A/B (TMT A/B). β denotes standardized regression coefficient. ^a^ Model 1, hsCRP (TMT A: F = 3.20, df = 1/340; *p* = 0.074; TMT B: F = 6.34, df = 1/340; *p* = 0.012), ^b^ Model 2, hsCRP adjusted for age, sex, illness duration, verbal intelligence quotient (IQ), and Beck Depression Inventory-2 (BDI-2) (TMT A: F = 10.65, df = 6/335; *p* < 0.001; TMT B: F = 14.42, df = 6/335; *p* < 0.001). ^c^ Model 3, hsCRP adjusted for variables in Model 2 and body mass index (BMI), diastolic blood pressure (DBP), triglycerides, history of diabetes, high-density lipoprotein (HDL) (TMT A: F = 6.55, df = 11/330; *p* < 0.001; TMT B: F = 8.17, df = 11/330; *p* < 0.001). * Significant by using the false discovery rate.

**Table 4 nutrients-12-03019-t004:** The association of serum hsCRP concentration with Stroop Test performance as a function of age, verbal IQ (*N* = 342).

	Stroop Word Reading			Stroop Color Naming			Stroop Interference		
	β	*t*	*p*	β	*t*	*p*	β	*t*	*p*
hsCRP ^a^	0.08	1.62	0.107	0.07	1.21	0.229	0.08	1.4	0.162
hsCRP ^b^	0.06	1.22	0.224	0.04	0.82	0.412	0.04	1.44	0.413
Age	0.21	−0.26	<0.001 *	0.17	3.01	0.003 *	0.25	4.77	<0.001 *
IQ	−0.14	−2.62	0.009 *	−0.15	−2.79	0.006 *	−0.27	−4.71	<0.001 *
Sex	0.02	0.27	0.787	0.09	1.61	0.107	0.07	1.6	0.18
Illness duration	−0.11	−2.06	0.04	−0.06	−1.03	0.303	−0.12	−2.35	0.029
BDI−2	0.11	1.98	0.049	0.12	2.18	0.03	0.2	3.65	<0.001 *
hsCRP ^c^	0.09	1.45	0.146	−0.00	−0.01	0.989	−0.01	0.16	0.371
Age	0.22	3.87	<0.001 *	0.17	3.04	0.003 *	0.25	4.62	<0.001 *
IQ	−0.14	−2.53	0.012	−0.14	−2.54	0.012	−0.26	−4.52	<0.001 *
Sex	0.04	0.653	0.514	0.08	1.25	0.211	0.02	0.95	0.501
Illness duration	−0.11	−2.00	0.047	−0.05	−0.91	0.366	−0.12	−2.26	0.037
BDI−2	0.12	2.13	0.034	0.12	2.17	0.031	0.19	3.48	<0.001 *
BMI	−0.00	−0.00	0.972	0.11	1.69	0.093	0.15	1.74	0.012
DBP	−0.04	−0.04	0.537	0.02	0.38	0.704	0.08	1.1	0.155
Triglycerides	−0.04	−0.04	0.593	−0.07	−1.02	0.311	−0.01	−0.46	0.878
HDL	0.01	0.01	0.935	−0.05	−0.64	0.521	−0.06	−0.51	0.613
Diabetes	−0.07	−0.07	0.181	−0.10	−1.79	0.074	−0.06	−1.09	0.278

Note: Data of linear regression analyses for high-sensitive c-reactive protein (hsCRP) and subtests of the Color–Word Interference Test after Stroop. β denotes standardized regression coefficient. ^a^ Model 1, hsCRP (Stroop word reading: F = 2.91, df = 1/341; *p* = 0.107; Stroop color naming: F = 1.45, df = 1/341; *p* = 0.229; Stroop interference: F = 1.97, df = 1/340; *p* = 0.162), ^b^ Model 2, hsCRP adjusted for age, sex, illness duration, verbal IQ, and Beck Depression Inventory-2 (BDI-2) (Stroop word reading: F = 4.37, df = 1/341; *p* < 0.001; Stroop color naming: F = 3.87, df = 6/341; *p* = 0.001; Stroop interference: F = 10.54, df = 6/340; *p* < 0.001). ^c^ Model 3, hsCRP adjusted for variables in Model 2 and body mass index (BMI), diastolic blood pressure (DBP), triglycerides, history of diabetes, high-density lipoprotein (HDL) (Stroop word reading: F = 2.67, df = 6/341; *p* = 0.003; Stroop color naming: F = 3.87, df = 11/341; *p* = 0.002; Stroop interference: F = 6.96, df = 11/340; *p* < 0.001). * Significant by using the false discovery rate.

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
