# Peer review of "C-Reactive Protein as a Possible Predictor of Trail-Making Performance in Individuals with Psychiatric Disorders"

_nutrients, 2020, doi:10.3390/nu12103019_

Round 1
Reviewer 1 Report
I commend the authors on their enthusiasm with which my and other reviewers' comments were addressed. Unfortunately, I believe more work is to be done.
First, I strongly recommend a thorough review of the text to improve readability. In part, readability could be improved just by having someone with a solid grasp of the English language edit it. In addition, the extensive rewrite seems to have led to some strangely formed sentences that need to be rewritten.
More seriously, there are also many inconsistencies in what is considered the main aim of the manuscript. At first glance, it seems the authors have decided to focus solely on CRP and metabolites of the TRP/KYN pathway. However, they repeatedly refer to obesity-related parameters not just as confounders, but as predictors of interest, both in the abstract and in the discussion. This all needs to be harmonized so that the manuscript represents one main aim.
On a more indepth-level:
Discussion, 2nd paragrap: in contrast to what the authors state here, the size of the association between hsCRP and TMT-B does not change much when covariates are added (from .14 to .10-.12). I think the authors refer to the p-value, which does change and is not significant when adjusting for multiple tests. This however has nothing to do with the effect size. Statements regarding this (first few sentences) need to be revised.
KYN as immune inflammation mediator: I appreciate that the authors took note of my previous comment and now do not refer to KYN or its metabolites as inflammatory. However, "immune inflammation" seems a little excessive; I would go for inflammatory mediator.
I'm curious: the authors state in their introduction that the novelty of their project is at least in part due to controlling for obesity-related confounders. They then find that while CRP is associated with TMT-B, this is not the case anymore after controlling for these confounders. What do the authors now conclude? Were previously found associations maybe incorrect because these confounders were not taking into account?
Author Response
Dear Reviewer, thank you very much for your careful revision of this manuscript. We agree with all your comments and have incorporated your suggestions in a revised version. We took head of your advice and double-checked English language– first by our co-author and colleague from Canada Prof. Roger McIntyre, second by our scientific assistant Valerie, living in UK.
In addition, the whole manuscript was revised regarding readability, the order of some passages were changed.

Reviewer 2 Report
The current review article by Nina Dalkner entitled 'C-reactive protein as a possible predictor of trail-making performance in individuals with psychiatric disorders’ is a very carefully written review. Their effort is appreciated and the manuscript in its original form has much strength. I have carefully reviewed the content and have no comments.
Author Response
Dear Reviewer, thank you very much for your careful revision of this manuscript. We agree with all your comments and have incorporated your suggestions in a revised version. We took head of your advice and double-checked English language– first by our co-author and colleague from Canada Prof. Roger McIntyre, second by our scientific assistant Valerie, living in UK.
In addition, the whole manuscript was revised regarding readability, the order of some passages was changed.

Reviewer 3 Report
Low-grade Inflammation and obesity may be associated cognitive dysfunction in psychiatric disorders. The authors investigated the C-reactive protein levels and the kynurenine-Tryptophan ratio on cognitive function in 364 patients. They report that of C-reactive protein levels are associated peripheral inflammatory marker in cognitive flexibility and psychomotor processing speed in psychiatric illness.
The data extend previous studies that low-grade Inflammation is associated with cognitive dysfunction.
Comment:
- C-reactive protein may represent a peripheral inflammatory marker in cognitive flexibility and psychomotor processing, but is really not specific for cognitive dysfunction. Additional serum biomarker might be of interest such as IL-1, IL-6 and more.
- Graphical presentation of the correlations of C-reactive protein in the different clinical tests would be easier to grasp.
Author Response

(The authors gave the same response as above.)

Round 2
Reviewer 1 Report
The authors have sufficiently addressed my comments.
This manuscript is a resubmission of an earlier submission. The following is a list of the peer review reports and author responses from that submission.
Round 1
Reviewer 1 Report
An observational study including 364 patients with diverse psychiatric disorders. Patients were tested for cognitive functioning using a Stroop test and the Trail Making Test A/B. Performance on the test was then associated with a wide range of biomarkers. The only significant result was an association between hsCRP and TMT B performance.
While the sample size is impressive and the authors assess an assay of biomarkers relevant both for cognitive function and for psychiatric disorders, I have several major concerns.
Overall, the manuscript could be improved in readability. The introduction seems to jump from topic to topic without a clear idea in mind. It’s also not clear from the introduction what is new about the presented study: associations between inflammation/tryptophan metabolites and cognitive function have been reported on before. The results section needs more detail: many tests results are not provided, making it difficult to ascertain what exactly was done and what the relevance is. Most work on inflammation and trypthophan metabolites with cognitive function in psychiatric patients seems to focus on mood disorder, particularly depressive disorder. The sample presented in this manuscript seems a mix of mood, stress, and somatoform disorders. It should be explained why the authors expect associations to be similar across these disorders.
The lack of a control group is an issue: without it, it remains unclear whether the findings are specific to psychiatric patients or not.
Pertaining to the tests of cognitive function: the Stroop test is not a standard test for assessing cognitive function but was designed for experimental purposes. This is not to say it can’t be used, but the authors should be clear about this. The authors mention they used norm values “from the manual”. As far as I know, there is no manual for the Stroop test. I would be interested to see where they got the norm scores from. In addition, when interpreting the Stroop test, it is common to use the difference between congruent and incongruent trials as the main outcome (thereby controlling for example for general reaction time). I guess this was not mentioned in the manual.
About the biomarkers: The authors keep referring to the markers from the tryptophan/kynurenine pathway as “inflammatory”. This is incorrect. While it is true that IDO is the main instigator of breaking down tryptophan and the metabolites are therefore often the result of increased inflammation, this does not make them “inflammatory”.
Analyses: From what I can make out, all biomarkers were included together in each of the three regression models. Considering that at least the TRP and KYN markers tend to be strongly associated, there’s a high risk for multicollinearity in the models. This should be checked and if present, remedied.
A more clear account of what variables were included in the models (and why) is needed. What are the predictor variables of interest and which ones are included as covariates (and why).
When presenting the regression models, all entered variables should be presented. Not just the ones that were significant.
Reviewer 2 Report
This paper is an interesting look at the relationship between inflammation (as measured by high-sensitivity CRP) and cognitive function (as measured by Stroop word/color tests and Trail-making tests). Multiple linear regression models were used to explore this relationship, adjusting for demographics and metabolic factors. The main findings are that hsCRP was associated with performance on the Trail making test B, which measures cognitive flexibility and executive functioning. Other relationships were found but they did not survive Bonferroni correction.
I believe that the paper is a valuable contribution to what is known about inflammation and cognitive function, particularly as the authors include multiple measures of the latter (i.e., Stroop and Trail making tests). However, there are some areas where I feel the manuscript could be improved.
First, the paper seems to focus heavily on the role of obesity, for example throughout the introduction and discussion, but there are instances where the importance of obesity per se seems to be forgotten. For instance, the stated aim of the study is to “investigate the potential role of inflammatory processes and metabolic parameters in cognitive functioning in a large sample…(p. 3, lns 105-106),” but the Discussion says that “the purpose of this study was to explore the potential role of obesity-related parameters and inflammatory processes on attention and executive function…(p. 7, lns 257-258).” Similarly, the title makes no mention of obesity or metabolic factors.
Certainly, obesity and metabolic factors are related but obesity can be considered a clinical condition. If the focus is truly on obesity, I would suggest that the authors consistently stress this in the manuscript. At the same time, if the authors are interested in determining the role of inflammation on cognitive functioning in obese individuals (who are assumed to have a higher inflammatory burden: p. 2, lns 48-49) vs. non-obese individuals, I would suggest that their analyses account for clinical obesity (28% of their sample) rather than BMI as a continuous variable. They could consider including a dummy variable for obese/non-obese in their regressions or possibly an interaction term between CRP and obese/non-obese to explore these differences.
On the other hand, if obesity per se is not a main focus, the authors should clarify this throughout the manuscript.
I think the Introduction could be improved by clarifying the processes connecting mood disorders, overweight/obesity, inflammation, and cognitive functioning. What is the directionality (or multiple directionalities)? It seems that cognitive functioning could lead to problems with weight then to an increased inflammatory burden and further downstream problems with mood and cognitive functioning (p. 2, lns 77-80), but weight problems might also precede problems with cognitive functioning. I’m sure there are many possible pathways and complex interactions, but outlining some of this in the introduction will, I think, help readers have a sense of where the authors’ data and results fit and which areas are best for clinical or public health interventions.
I’m also a little confused by the “premorbid” verbal IQ measure. It is called “premorbid” (presumably pre-diagnosis of psychiatric disorder?) but on p. 3 lns 137-139 it sounds like the verbal IQ test was administered at the same time as the rest of the cognitive tests. How can this be considered premorbid?
Similarly, I’m wondering about the TMT A and B description on p. 4, lns 148-151. Part B asks participants to connect numbers and letters in sequence and alternating. Part A connects numbers only “which does not require shifting (ln 149).” However, the next sentence says that part A assesses attentional shifting. Perhaps this is a typo?
Please remove the sentence about R squared (p. 6, lns 218-220). This is fairly common knowledge.
In the Clinical Impact section, the authors state that “we also know obesity and the investigated metabolic parameters (BMI…) seem to play a tangential role in the prediction of attention…(p. 8, lns 322-324).” But none of these parameters were statistically significant in the authors’ models, even prior to Bonferroni correction. Please clarify or remove.